# University Dropout Model for Developing Countries: A Colombian Context Approach

**DOI:** 10.3390/bs13050382

**Published:** 2023-05-05

**Authors:** Alejandro Valencia-Arias, Salim Chalela, Marcela Cadavid-Orrego, Ada Gallegos, Martha Benjumea-Arias, David Yeret Rodríguez-Salazar

**Affiliations:** 1Escuela de Ingeniería Industrial, Universidad Señor de Sipán, Chiclayo 14001, Peru; 2Dirección de Transferencia Tecnológica y Propiedad Intelectual, Universidad Señor de Sipán, Chiclayo 14001, Peru; dasalazar@crece.uss.edu.pe; 3Research and Innovation Office, Universidad del Rosario, Bogotá 111321, Colombia; salim.chalela@urosario.edu.co; 4Observatorio DRAF—Oficina Asesora de Planeación—Inder Alcaldía de Medellín, Medellín 0500304, Colombia; marcela.cadavid@inder.gov.co; 5Instituto de Investigación y Estudios de la Mujer, Universidad Ricardo Palma, Santiago de Surco 15039, Peru; ada.gallegos@urp.edu.pe; 6Facultad de Ciencias Económicas y Administrativas, Instituto Tecnológico Metropolitano, Medellín 050034, Colombia; marthabenjumea@itm.edu.co

**Keywords:** dropout rate, higher education, university students, retention

## Abstract

University dropout is a problem that has raised great concern in institutions of higher education. For this reason, academic institutions need to study the phenomenon and come up with alternatives that contribute to the improvement of students’ individual commitments. The aim is to examine the dimensions influencing the decision of university students to drop out. A quantitative approach study was carried out, based on a cross-sectional exploratory-descriptive field design, in which 372 students participated. According to the participants, one of the dimensions that influence the decision to leave the university is the support of the institutions to the continuity of the motivation processes to the student body, since the easy access to the credits is greater than the scholarships granted, which coincides with the financing restrictions of university students in developing countries. In conclusion, it is observed that the communication between managers, teachers, and students is a key factor in the processes of academic retention as a strategy to combat the phenomenon of university desertion.

## 1. Introduction

University dropout is a problem that has raised great concern among the governing bodies of higher education institutions in recent decades due to its far reaching implications [1,2]. These implications are evident, on the one hand, in financial terms, when there is instability in the source of student income [3], and on the other hand, in terms of the students questioning the efficiency of the higher education system, as only a fraction of students who begin their undergraduate studies complete them [4].

This issue has become one of the most controversial issues when assessing the efficiency of the university system and the accountability of educational policy [5,6]. For this reason, academic institutions need to study the phenomenon and come up with alternatives that contribute to the improvement of students’ individual commitments, addressing the dimensions that are strongly linked with student dropout within higher education institutions [7].

Higher education has always been a fundamental objective of the different cultural elites. A few years ago, access to education was reserved for a few, excluding numerous groups, for example, women, people with few economic resources, or people with disabilities. In today’s knowledge economy, global enrollment in higher education has increased dramatically in different regions, such as Europe, Asia, the Arab States, Sub-Saharan Africa, Small Island Developing States, North America, Latin America, and the Caribbean [8].

Both states and universities have taken initiatives to increase access to education among low-income groups, as well as racial and cultural minorities, after the 2009 UNESCO World Conference on Higher Education recognized higher education as a “public good” [9]. Similarly, the United Nations 2030 Agenda for Sustainable Development commits to ending poverty, fighting inequality and injustice, and addressing climate change. In this instance, SDG 4 refers to inclusive education, which is expected to ensure access to education and substantially increase the number of youth and adults who have the necessary skills, particularly technical and professional skills, to access employment, decent work, and entrepreneurship [10].

It is noteworthy that this phenomenon has been studied from different approaches, including social pressure, family environment, university context, and psychosocial aspects linked to the individual [11,12]. Therefore, many studies insist on acknowledging the differences between dropout causes, as it is possible to skew institutional initiatives or strategies and take a wrong approach to solutions, as these should be consistent with the dropout risk situation, with differential impacts in various socio-cultural contexts [13,14]. In addition, among the dropout reasons declared by students, it has been found that there is a rupture between dropout and academic difficulties in developed countries such as Spain and Portugal, where it has been identified that the dropout may be more related to the student’s personal and psychological dimensions [5,6,7,8,9,10,11,12,13,14,15].

Currently, there are also key categories that encompass the variables that have the greatest effect on the process of university dropout: (i) individual aspects, covering aspects such as age, gender, family group, and social integration; (ii) academic aspects, including items such as vocational guidance, intellectual development, academic performance, study methods, admission processes, degrees of satisfaction with the program, and academic load; (iii) institutional aspects, in which elements such as academic regulations, student funding, university resources, quality of the program, and relationships with teachers and peers are considered; and (iv) socioeconomic dimensions such as socioeconomic stratum, the student’s employment status, the employment status of the students’ parents, and the parents’ educational level [16].

In developing countries, students with low income have found restrictions to obtain loans or funding for their university majors to be an obstacle to continuing their academic careers [17]. In the case of Colombia, the university dropout rate is 54.34% [18] for the 11 cohort of 2017. Research has focused on the existing correlation between financial aid offered to students and dropout rates, concluding that there is a 25–29% decrease in the probability of dropping out among students who receive financial aid. On the other hand, this aid was noted to be more effective at the beginning of university courses, as this is one of the appropriate strategies to improve the retention rates in higher education in the country [19]. Under this scenario, the dropout intention in Colombia is a concern that is both academic and political. According to [18], university dropout does not result from a single cause, but reflects multiple dimensions of an academic, financial, socio-economic, institutional, and vocational nature.

Therefore, the study of school dropout in developing countries, and its subsequent follow-up, is very important, as this phenomenon is regarded as an indicator of quality in university management. In fact, its valuation is an indicator in many models of evaluation of academic institutions, and in turn, it is an item used in terms of ranking the best universities and those that can gain access to a high-quality accreditation [20]. Therefore, the aim of this study is to examine the dimensions influencing the decision of university students to drop out.

### Theoretical Framework

Globally, various theoretical models for student dropout have been proposed. Table 1 describes the main existing models, according to their representatives, year of creation, and main characteristics. This will provide an overview of the different methods of understanding the phenomenon of student dropout. In each of them, five categories are identified to classify approaches to dropout. Psychological, sociological, economic, organizational, and interactionist aspects were proposed by the authors of [21,22] to classify these models.

Despite the relevance of the models considered in this context, it is difficult to ex-plain dropout based on a specific type of variable, so when some researchers have attempted to replicate their theories, they have stumbled upon the multidimensional nature of this phenomenon. Such is the case for ref. [24], which, based on Tinto’s model, found social origin and student performance to be key dimensions for dropout. Many studies have discovered no difference between the levels of dropout among groups of students who have failed and those who have passed, and in other situations, it has even been noted that students that drop out have higher academic grades than those who persist, so it must be pointed out that there is no comprehensive explanatory model that includes the wide range of circumstances surrounding this phenomenon [25].

To clarify the methodological path, the authors have chosen to take the model proposed by [26] as a reference axis for the development of this study. Spady’s sociological perspective enables the understanding of university student dropout by the confluence of internal and external dimensions, as can be seen on Figure 1.

The model addresses important dimensions that, according to the literature, are applicable to the Colombian context, with successful results. Family background can lead people who have begun higher education programs to abandon them. In this regard, research by the authors of [27,28] showed family factors as the third-leading cause of university dropout in an investigation conducted with food engineering students at Universidad de Cartagena.

Social integration, in the context of developing countries, can be seen as a variable that significantly influences a student’s decision to drop out. In line with the foregoing findings, [29] it is proposed that students’ interpersonal relationships can prevent their proper connection with a new context, such as university [30].

Thus, there is literature that assesses different dimensions (psychological, financial, sociological, organizational, and interactional) in the relationship between students and HEIs, and it all concurs that university dropout is an inherently individual decision [31]. Thus, HEIs, irrespective of their legal nature, are increasingly concerned about implementing strategies to mitigate this phenomenon, not only due to the consequences it brings for the financial structures of these institutions, but also due to the impact on social order, significantly compromising the development and growth of the communities [32].

## 2. Materials and Methods

### 2.1. Research Model

First, it is important to mention that the model to be validated is based on the one proposed by ref. [26], but it presents a set of relationships that are new or that have had an incipient development in the Colombian university system, which the authors suppose are in accordance with the cultural, economic, and social environment of undergraduate students. These include: family background; conditions of social integration; institutional commitment, and peer support associated with the decision to drop out (See Figure 2).

The aim of this study is to examine the dimensions influencing the decision of university students to drop out of. This research is based on a cross-sectional exploratory-descriptive field design using a quantitative methodological design, proposing a self-administered questionnaire as an instrument for collecting information. For the construction of the questionnaire, studies published in international indexed journals by the most renowned authors on the subject were consulted. Based on this, a database was built with 75 questions that mediate in different aspects, from psychological, sociological, economic, organizational, and interactionist approaches. Based on the needs of the study and the criteria of the research team, 37 questions were selected, grouped into 9 dimensions, including: family background, policy coherence, social integration, student satisfaction, peer support, institutional commitment, intellectual development, academic performance and academic potential, and decision to drop out.

To choose the questions, the fact that they had been validated in prior studies in countries with a cultural, economic, and social context similar to those of Colombia was considered to ensure that the dimension–variable relationships were correctly established. Questions posed using a Likert scale were preferred in order to facilitate the analysis of the results, as this is the most widely used scale and provides the greatest diversity of information to validate this type of model [16,17,18,19,20,21,22,23,24,25,26,27,28,29,30,31,32,33,34]. The nine dimensions are defined as follows:

Decision to Drop Out (DDO): This dimension is associated with the variables related to the student’s decision to drop out. The reasons evaluated correspond to aspects such as family problems, not having financial support, the lack of vocational and professional guidance, the differences in the academic requirements between the school and the university, the number of credits that must be met to complete a major, the difficulty to meet the payment of the tuition, the difficulty to meet additional costs, and psychological factors.

According to [35], the pathway model suggests that the decision to drop out can lead to changes in students’ attitudes, interests, goals, or motivation, which in turn, will have positive or negative effects in later stages of the university career. This is ultimate outcome, so this dimension is directly affected by academic performance and institutional commitment.

Student Satisfaction (SS): The variables grouped under this dimension are those that assess student satisfaction with the services provided by the university in general. Each item measures the students’ perception of the relevance and significance of the academic program, as well as the price and financing options compared to other institutions. This influences demotivation due to expectations of low income and unemployment in the future, which can be a major reason for not continuing studies in higher education.

Policy Coherence (PC): The variables contained in this component refer to policy coherence, i.e., what students experience regarding the decision to terminate their higher education studies, in terms of their environment. This component assesses items relating to how interesting and fun the profession can be for students, as well as the comfort students feel when studying for a career and the satisfaction expected from fulfilling these expectations. Therefore, this dimension addresses the social adaptability of students with their peers and the conflicts generated between teachers and/or students.

Social Integration (SI): This dimension relates to items based on the students’ affinity with the university environment. In this way, people turn out to be a determining factor in the motivation to complete university studies.

Family background (FB): This is associated with the variables of support received by the students’ families, personal or family obligations that are more time-intensive for the person completing their major; namely, responsibilities related to fatherhood, motherhood, or any other kinship that does not allow the student to devote him or herself entirely to his or her undergraduate studies. This dimension also includes difficulties that students experience in completing their university studies, such as pregnancy, the need to care for children, and whether or not their parents have completed university studies.

Institutional Commitment (IC): This component is associated with the variables related to satisfaction with the academic service offered by the institution. The services evaluated correspond to aspects such as: the infrastructure or location of the university that may lead students to abandon their university studies, even though they enjoy them, the lack of knowledge regarding student benefits and university welfare, and the denial of a scholarship application by the university.

Peer Support (PS): This dimension includes the variables responsible for measuring student satisfaction regarding professors’ teaching methodology. Each variable is responsible for measuring student perception regarding the main motivators to continue pursuing university studies, as well as assessing aspects related to student permanence in the university. In this way, the teaching methodology of the tutors, the relationship with the tutors, and the lack of vocational and professional orientation by their teachers are considered.

Academic Performance (AP): This comprises two items related to the ease and/or difficulty experienced by students regarding the development of their professional careers, based on two fundamental pillars: on the one hand, the schedules of the subjects offered by the dean’s office and the office of the academic vice-chancellor, and on the other, the interest, perseverance, and self-discipline of students to address the academic commitments set out by these two offices. This is related to psychological problems, such as depression, affective or family instability, and even the change of marital status, that can influence academic performance.

Intellectual Development (ID): This component is associated with the variables related to the fulfillment of the higher education service provided by the institution, measuring how the job supply (demotivation due to expectations of low income and unemployment in the future), the relevance of the courses, and the access of parents regarding the implementation of university studies influence the decision to complete a major. In this dimension, academic demand is very important, as well as the rigorousness of the academic program and the number of credits required to complete the academic program.

### 2.2. Participants

A non-probabilistic sampling by criterion was used, seeking that the population studied corresponded to Colombian higher education students. The sample featured a total of 372 university students enrolled in undergraduate programs, coming from a middle-low socioeconomic stratum. Prior to the collection of information, a pilot test was conducted with 45 students in order to assess the clarity and understanding of the structure and questions of the questionnaire. Subsequently, the tool’s final version was adjusted and applied to the selected population, and the questionnaires were checked for quality, with 357 questionnaires determined suitable for data analysis.

The ethics committee of CE-CIES approved the study, with approval code (ACTA13072020). Additionally, informed consent was obtained from all participants prior to completing the survey.

### 2.3. Procedure

Since the initial goal was the implementation and validation of a student dropout model based on the review and validation of the main risk dimensions associated with this phenomenon, proposing relationships adjusted to the Colombian context, the first task consisted of validating the model through an exploratory factor analysis. The statistical treatment was carried out with the support of IBM SPSS Statistics software, version 25, for Windows.

Next, for the validation of the variables to be studied the model, which are concentrated in the dimensions family background; peer support; institutional commitment; policy coherence; academic performance; decision to drop out; intellectual development; social integration, and student satisfaction, all those that are defined as predictors or are related to the student’s perception regarding their major and the expected quality of the program, as explained in the theoretical model, were selected.

For the exploratory factor analysis, the aim is to determine that the variables are correctly correlated with each of the dimensions. An initial extraction phase is performed with the varimax rotation approach, a method related to orthogonal rotation, used to minimize the number of variables that have high loadings on each dimension [36]. Model estimation is also performed using principal component analysis to group the variables into a few unrelated factors. In this way, an initial correlation analysis is performed by identifying the value of the factor loadings [37].

It was then found that the 37 variables analyzed are grouped into nine dimensions, as presented in Table 2. In this first phase of the analysis, it was observed that the data collected in the research does not show redundant information, since the factor loadings of the observable variables were mostly above 0.6, making it possible for the average factor loadings for each dimension to obtain a value exceeding 0.7 in order to achieve the model’s convergence [38]. 

## 3. Results

To analyze the validity of the model, an evaluation based on convergent and discriminant validity is performed. The statistical indicators of the Kaiser–Meyer–Olkin index (KMO) and Bartlett’s test of sphericity are used, establishing that values equal to or greater than 0.5 for KMO are acceptable, as well as values of 0.000 for Bartlett’s test, as a recommendation to continue with the factor analysis [38]. The discriminant validity analysis is performed using 95% confidence intervals using Fisher’s method proposed in ref. [39], which establishes that there is convergent validity when the value 1 is not obtained, a criterion that is also met.

On the other hand, the reliability of the model is also validated, i.e., which variables are consistent with the dimensions according to the measurement scale. For this, the Cronbach’s alpha statistic is used, establishing that it is necessary for each dimension to have a value equal to or greater than 0.7 [40], which is also fulfilled.

The estimation of the proposed structural model for the identification of the most influential dimensions in the decision to drop out was conducted (see Table 3), gathering the various hypotheses proposed and measuring their degree of association by means of the Somers’ D statistic, which corresponds to a measure of association between two ordinal variables [41].

From the results of Figure 3, it is noted that the results obtained from these hypothetical relationships show that aspects related to family background have a significant correlation with the decision to drop out (0.402), showing that the family unit has an important influence in the university environment, as students involuntarily or voluntarily respond to stimuli received from the authority figures that surround them. This can generate a good experience in the student, in turn becoming the key tool to change or improve the perception of the student body regarding the institution and its leaders, since it is here that they can fill different needs and increase their level of satisfaction with the service received. Similarly, the views of external actors (such as friends) has a strong relationship with the student’s satisfaction and feeling of comfort regarding the major and the institution where he or she is pursuing it (0.502). Social integration is strongly related to the decision to drop out (0.405), thus confirming the importance of experiences lived within the environment, given that the context in which the student moves become a fundamental strategic asset for institutions, marking the way towards quality of service and differentiation.

On the other hand, intellectual development, peer support, and academic performance reflect a high level of association with the decision to drop out dimension, which may indicate that teacher assistance services not only allow students to interact with their mediators and receive personalized attention, but also provide a link to the institution for undergraduate or postgraduate training programs to be addressed with a greater sense of appropriation by students.

As a complement to the model, it was noted that most of the people surveyed (82%) have not taken school leave for any semester of their undergraduate program, despite the fact that more than 45% currently have a job, which corroborates that time-intensive personal or family obligations are not a dimension for young people to postpone their major (23.81%), so it can be established that the attitude toward behavior has a remarkable influence, not only in behavioral intention, but in the subject’s behavior in general. Additionally, participants agreed that the lack of vocational and professional guidance offered by their teachers is a cause to leave their higher education studies (29.13%), and affinity with the university environment and people is a factor that motivates students to finish their higher education studies (37.08%), which leads the authors to believe that the dropout behavior is not due to cultural practices, as much as to the subject’s sole desire to carry out the behavior.

## 4. Discussion

The interpretation made by the authors of [42] of the student dropout phenomenon in the university arena calls for a two-way analysis in which not only the individual’s endogenous aspects can be assessed, but also their belonging in an ecosystem that directly impacts their motivations, and hence, their behavior.

The study conducted by the authors of [43] shows that personal and family problems are one of the main reasons why students do not renew their enrollment in the institution. Consistent with these studies in the country, other approximations in nations of the region, such as Chile, show the impact that family has on university dropout: in light of the findings made by ref. [44], along with educational credits, families are the main source of financing of higher education studies, so when faced with difficulties at home, the likelihood of leaving or postponing education is high. This conclusion was also validated by ref. [45].

Additionally, ref. [46] mentions the various studies that have been conducted in Latin America, in which, consistent with one of the findings of the investigation presented in this article, there is a positive association between the completion of professional studies and having university graduate parents, as well as their type of occupation.

On the other hand, ref. [47] indicates that an environment that is unfavorable toward individual stability, such as one in which there is conflict with teachers and classmates, may influence the student’s decision. Moreover, in Chile, it was found by ref. [48] that in light of social representations, facts such as migration for advancing university studies be-come a factor that affects the social and emotional stability of students. This validates the results of the research presented, but it also provides new clues about future research that may be implemented in the field.

However, in this study, it was not evident in the Colombian context that institutional commitment is a factor associated with students’ decision to drop out. This is in line with the findings made by ref. [43] at Universidad de Caldas (Colombia), in which they confirm that one of the factors with the lowest impact on the decision to abandon a program is, indeed, institutional administrative matters.

Likewise, the development of this study suggests the impact of peer support—including aspects such as teaching methodology and non-flexible programs—have on permanence in education and university dropout. Some explanations for this relationship have been expressed earlier by the importance given by students to the perception of the academic quality of their teachers as a starting point to continue or suspend their university education [49]. In Peru, it has already been indicated that poor pedagogical and teaching practices that do not allow for the development of coordinated work with peers are a triggering factor for dropout.

Similarly, as a missionary axis, universities must develop strategies to mitigate dropout. Thus, the Ministry of National Education (MEN) and the National Accreditation Council (CAN) have proposed that institutional welfare departments work on this goal through programs such as: parents’ schools, academic strengthening activities (math tutoring clinics, workshops on study techniques, job market insertion seminars, vocational guidance workshops, and psychological support for students on academic probation or coming back to school after having dropped out, among other activities), workshops that strengthen the human development of students who are at risk for becoming dropouts, committees for the prevention of psychoactive substance consumption, as well as sports, recreational, and cultural activities.

## 5. Conclusions

The scale of the university dropout phenomenon is undoubtedly a thematic priority in the political agendas of most Latin American countries. As could be seen throughout this article, this is a problem that touches the social and economic aspects of society, compromising their growth and development. Thus, the complexity of the university dropout phenomenon is worth highlighting, because even though the challenges are similar for Latin American countries, the motivations vary from one context to another.

University dropout is motivated by factors such as individual problems regarding social adaptation to new environments and family support, rather than by matters related to the institutional structure within HEIs.

In this sense and based on the most significant results of this research, it may be inferred that university dropout is motivated by factors such as student problems regarding social adaptation to new environments and family support rather than by actual institutional matters within HEIs such as flexible hours, length of the program chosen, and student services and benefits.

Regarding this last aspect related to institutional commitments, in Colombia and in different nations of the Southern Cone, the efforts to mitigate the effects of dropout are continuous, and currently, the strengthening of student welfare areas is a two-way goal, aiming at fulfilling the social function of universities to ensure education, but also as a strategy for the development of the institutions themselves (particularly those seeking high quality accreditation). Moreover, virtualization has been one of the strategies implemented by universities for a number of years, not only to expand the coverage of their academic offerings, but also to generate new pedagogical and teaching practices, such as distributive education and distance education. However, it is worth mentioning that the first results of studies on the effects of these new practices on student retention are only just beginning to surface.

The study was also able to confirm Spady’s theory, which is becoming a fundamental tool for evaluating and identifying the factors, variables, and relationships that inhibit or motivate a student’s decision to drop out; as seen in this model, not only the positive or negative evaluation of the performance of the behavior by the individual come into play, but also the social pressure of performing or not performing that behavior.

Therefore, and given the complexity entailed by the studied phenomenon, it is necessary to motivate further research in the field to explore the causes of dropout from various angles, since, as the literature mentions, while the main attributable factors are associated with economic variables and situations happening in families, the impact of social integration and relationships established by students with their peers on the dropout phenomenon cannot be ignored. Therefore, the results of this study serve as input for universities in the design of student retention strategies by defining the main motivators of desertion in the Colombian context, as seen in the sample used.

Among the study’s limitations, one is the type of sampling used. It is necessary to study a larger sample to obtain more consistent results that allow for contrasting the findings of this and other studies. Another limitation is the population used; it was only carried out in the Colombian context, which opens up opportunities for applicability in other contexts.

Finally, this study not only creates new knowledge, but also reaffirms existing postulates. Additionally, as a starting point for future research, it shows that although university dropout has been extensively analyzed in Latin America, the differences that can surface between one context and the other necessarily lead to new studies, and even to thinking about ambitious proposals that involve the analysis of multiple contexts, i.e., comparisons between nations with shared socioeconomic and demographic characteristics: e.g., Chile, Argentina, and Uruguay, as well as Colombia, Ecuador, and Peru.

## Figures and Tables

**Figure 1 behavsci-13-00382-f001:**
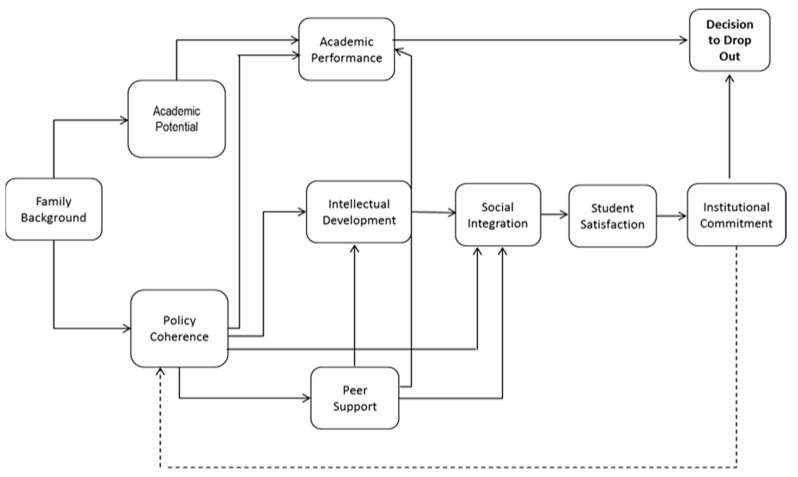
Spady’s model. Source: adapted from [26].

**Figure 2 behavsci-13-00382-f002:**
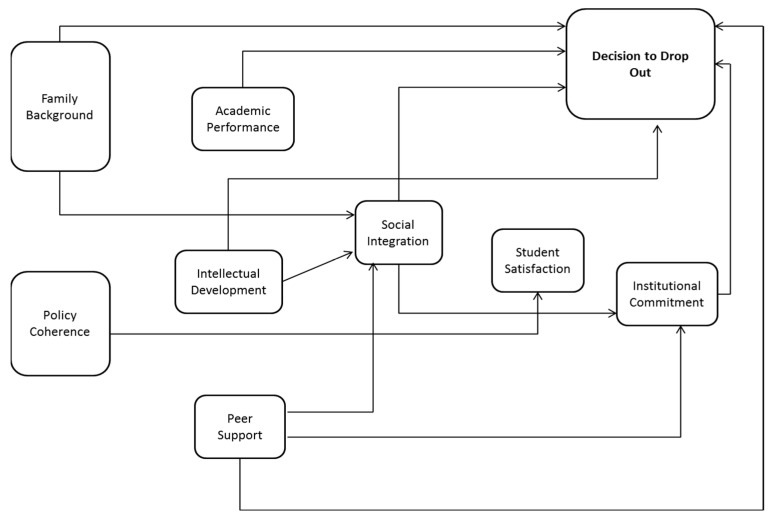
Model proposed for assessing the university student’s dropout intention. Source: own elaboration adapted from [26].

**Figure 3 behavsci-13-00382-f003:**
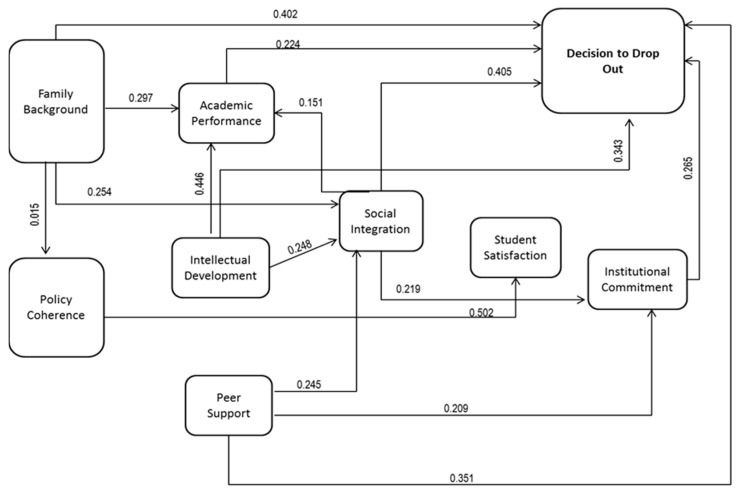
Model results. Source: own elaboration. Arrows indicate the null hypotheses that were validated in the research.

**Table 1 behavsci-13-00382-t001:** Major theoretical models for explaining university dropout. Source: own elaboration based on information adapted from [23].

Author	Year	Description
Spady	1970	This model is based on Durkheim’s theory of suicide, and it provides a guide focusing attention on the interaction between student attributes (i.e., dispositions, interests, attitudes, and skills) and the influences, expectations, and demands imposed by various sources in the university environment.
Fishbein and Ajzen	1975	This is one of the pioneer models of the psychological approaches, showing the influence exerted by beliefs and attitudes regarding behavior. Therefore, the intention of dropping out or going on in an academic program will be determined by prior behaviors and subjective norms.
Bean	1985	This model argues that university dropout is analogous to productivity and describes the importance of behavioral intentions (to stay or leave) as predictors of persistence. It is assumed that behavioral intentions are configured in a process in which beliefs form attitudes, and these, in turn, come into play when forming behavioral intentions. The model also claims that beliefs are influenced by components of the academic institution (quality of the courses and programs, teachers, and peers).
Pascarella and Terenzini	1985	This is a general causal model, with explicit considerations regarding institutional and environmental characteristics. The authors argue that the development and change in students occurs in five sets of variables: skills, yields, personality, aspirations, and ethnicity.
Tinto	1989	The model proposed by the author is interactionist. Tinto considers that as the student goes through high school, various variables contribute to reinforce his or her adaptation to the institution selected by him or her, as he or she enters it with a set of characteristics that influence his or her experience in post-secondary education. These include family background characteristics, such as the family’s socio-economic and cultural level and the values it sustains, as well as personal attributes and those related to the academic experience.
Ethington	1990	The author takes the legacy databases created by Fishbein and Ajzen (1975) as a reference for the development of this model. Ethington proposes a more general scheme based on achievement behaviors. In this model, preliminary academic performance (in which the self-concept of the student’s performance and the perception of the degree of difficulty of studies are key) and family background, coupled with a solid system of values, as well as expectations regarding success, are consolidated as guarantors of academic permanence.

**Table 2 behavsci-13-00382-t002:** Convergent validity of standardized factor loadings. Source: own elaboration supported by the SPS statistical software.

Dimensions	Variable	Standardized Factor Loadings	Average Number of Standardized Factor Loadings
Family Background	FB1	0.833	0.833
	FB2	0.833	
Institutional Commitment	IC1	0.473	0.704
	IC2	0.819	
	IC3	0.819	
Policy Coherence	PC1	0.796	0.796
	PC2	0.796	
Student Satisfaction	SS1	0.652	0.708
	SS2	0.816	
	SS3	0.657	
Decision to Drop Out	DDO1	0.795	0.701
	DDO2	0.618	
	DDO3	0.748	
	DDO4	0.559	
	DDO5	0.812	
	DDO6	0.799	
	DDO7	0.641	
	DDO8	0.615	
	DDO9	0.625	
	DDO10	0.666	
	DDO11	0.777	
	DDO12	0.761	
Social Integration	SI1	0.796	0.726
	SI2	0.731	
	SI3	0.652	
Academic Performance	AP1	0.717	0.717
	AP2	0.717	
Intellectual Development	ID1	0.736	0.736
	ID2	0.736	
Peer Support	PS1	0.749	0.749
	PS2	0.749	

**Table 3 behavsci-13-00382-t003:** Variable association analysis. Source: own elaboration.

	FB	IC	PC	SS	DDO	SI	AP	ID	PS
FB	1.000								
IC	0.107	1.000							
PC	0.031	0.092	1.000						
SS	0.016	0.181	0.502	1.000					
DDO	0.402	0.265	0.024	0.086	1.000				
SI	0.254	0.219	0.086	0.116	0.405	1.000			
AP	0.143	0.192	0.094	0.110	0.224	0.085	1.000		
ID	0.191	0.144	0.130	0.035	0.343	0.248	0.107	1.000	
PS	0.085	0.209	0.083	0.054	0.351	0.245	0.114	0.173	1.000

## Data Availability

The data that support the findings of this study are available from the corresponding author upon reasonable request.

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
