# Peer review of "University Dropout Model for Developing Countries: A Colombian Context Approach"

_behavsci, 2023, doi:10.3390/bs13050382_

Round 1

Reviewer 1 Report

Dear authors

Thank you for the manuscript presented and congratulations on the work done. The topic is actual and relevant to the international agenda for world development, and it is interesting to go further in the understanding of the phenomena in Latin-American contexts.

Unfortunately, several parts of the manuscript are not clear. It is necessary to revise the language of the manuscript, sentence structure, and some imprecisions in terms of ideas presented, but also, theoretical ones. I strongly believe that this is a central issue to address.

In this sense, I present some points to the authors' reflection in order to improve the overall quality of the manuscript.

For example, the use of the word “college”; is more adequate for the context in which the study was performed? Could maybe consider the use of “university” or higher education (indeed only used in the discussion section, abbreviation HEI); other examples will be presented by section.

Title and Abstract

Possibly, the title could be revised, it is really an approach that the authors want to present? Or an experience, case?  

The abstract could be improved and clarified; I suggest to do not starting the abstract with the aim of the study, some topic introduction will be interesting.

 Introduction

It will be interesting to reinforce the relevance of study phenomena in order to inform the institution with empirical data of its own context to inform decisions and contribute to intervention/prevention strategies and practices. Maybe reinforce the social responsibility and the value of having a degree for people from different countries/contexts; reinforcing the tertiary education as a tool to access a decent and fair job (for example, SDG).

1.1. This section presents relevant information and the main theoretical models; I suggest revising the models’ descriptions in order to synthesize and present all of them in the same way.

Methodology

It will be useful to create the subtopics of Participants, Instruments, and Procedures in order to allow the audience to better understand the information presented.

The use of the terms “construct”, “dimension”, “variable”, and “items” should be revised and clarified.

 In line 126 is presented the “nine constructs with the” – I believe this part should be strongly revised.

The expression “by its initials in Spanish” could be deleted and the acronyms or abbreviations should be adjusted, in order to ease the reading.

It is not necessary to create a subtopic for each construct, not even an instrument.

2.1 Decision to dropout – It is a construct, a variable,…? The description seems to be related to Dropout Motives.. pointed out by students? By theoretical models? Presented in instruments? which instruments? It is not clear.

2.2 The sentence is not finished. Student Satisfaction with “college unit” in terms of services or training, or staff, or all of them? Maybe using the terminology of the theoretical models could be useful, in order to connect with the general audience.

2.3. It is not clear if it is about students’ academic expectations or aspirations, parental/familial expectations,..

2.4 “Social Integration (…) assessing everything related to the commitment declared by the university with regard to its students.” Social integration? Or commitment? I really think that language issues do not allow to understand the main idea of the authors.

2.5. Family background in the literature about higher education dropout usually is presented in studies with first-year students, in several studies pointing to the relevance of the academic level of education of the mother and father. The academic level of the parents is associated with the socioeconomic characteristics of the families and in many countries associated with the support and the opportunities for academic and cultural exploration. For example, several studies presented higher rates of dropout in students with parents with lower levels of education. In the description the authors presented “responsibilities related to fatherhood, motherhood, or any other kinship that does not allow the student to devote him or herself entirely to his or her undergraduate studies” – this is also relevant to dropout but are not family background but Personal Variables, or Familiar Responsibilities.

The same problems exist for 2.6, 2.7, 2.8, and 2.9. The variables presented are not clear and are mentioned in a simple way without reflecting the complexity and interaction between variables (as mentioned by the authors), which several times have mediating or moderating effects.

 Procedures

It is not clear how the study occurs. Ethical procedures were not presented.

How data collection, data curation, and data analysis were made? Which software was used? 

 Results

The results are presented in a very short way and do not address the constructs mentioned.

 Discussion and Conclusion

In this section is presented new information that could be presented in the introduction in order to better know the Latin-American, or Colombian, contexts.

References

Could be improved with more references; it is also important to include the titles of the manuscript for the language of the article.

Author Response

February 22, 2023

Dear

Behavioral Sciences Editorial Office

Kind regards

According to the suggestions of our article by the reviewer, the following changes were made, properly marked with red letters in the article:

Reviewer

Comment

Response

R1

 the use of the word “college”; is more adequate for the context in which the study was performed? Could maybe consider the use of “university” or higher education (indeed only used in the discussion section, abbreviation HEI); other examples will be presented by section.

The word "university" is used instead of "College".

R1

Possibly, the title could be revised, it is really an approach that the authors want to present? Or an experience, case? 

The title is presented in a Colombian context.

R1

The abstract could be improved and clarified; I suggest doing not starting the abstract with the aim of the study, some topic introduction will be interesting.

The importance of acting against desertion by universities is included.

R1

It will be interesting to reinforce the relevance of study phenomena in order to inform the institution with empirical data of its own context to inform decisions and contribute to intervention/prevention strategies and practices. Maybe reinforce the social responsibility and the value of having a degree for people from different countries/contexts; reinforcing the tertiary education as a tool to access a decent and fair job (for example, SDG).

The information requested by the evaluator is included.

R1

Introduction. 1.1. This section presents relevant information and the main theoretical models; I suggest revising the models’ descriptions in order to synthesize and present all of them in the same way.

How the models are presented in Table 1 is unified according to the variables proposed and their importance.

R1

Methodology. It will be useful to create the subtopics of Participants, Instruments, and Procedures in order to allow the audience to better understand the information presented.

The methodology is separated into the following subsections: Research model, Participants, and Procedure.

R1

Methodology. The use of the terms “construct”, “dimension”, “variable”, and “items” should be revised and clarified.

The terms are unified in "dimension".

R1

Methodology.  In line 126 is presented the “nine constructs with the” – I believe this part should be strongly revised.

It is indicated that only the nine dimensions are explained.

R1

Methodology. The expression “by its initials in Spanish” could be deleted and the acronyms or abbreviations should be adjusted, in order to ease the reading.

The expressions "by its initials in Spanish" are deleted and the abbreviations are adjusted to English.

R1

Methodology. It is not necessary to create a subtopic for each construct, not even an instrument.

A subtopic is not used for the dimensions.

R1

Methodology. 2.1 Decision to dropout – It is a construct, a variable,…? The description seems to be related to Dropout Motives.. pointed out by students? By theoretical models? Presented in instruments? which instruments? It is not clear.

It is clarified that this is a dimension. Its explanation is expanded.

R1

Methodology. 2.2 The sentence is not finished. Student Satisfaction with “college unit” in terms of services or training, or staff, or all of them? Maybe using the terminology of the theoretical models could be useful, in order to connect with the general audience.

It is explained in what sense the student's satisfaction with the university is based. The terminology of the theoretical models is used.

R1

Methodology. 2.3. It is not clear if it is about students’ academic expectations or aspirations, parental/familial expectations,..

It is clarified that the expectations correspond to those of the students.

R1

Methodology. 2.4 “Social Integration (…) assessing everything related to the commitment declared by the university with regard to its students.” Social integration? Or commitment? I really think that language issues do not allow to understand the main idea of the authors.

A more accurate explanation of the dimension, which refers to the integration of students with other members of the university, is included.

R1

Methodology. 2.5. Family background in the literature about higher education dropout usually is presented in studies with first-year students, in several studies pointing to the relevance of the academic level of education of the mother and father. The academic level of the parents is associated with the socioeconomic characteristics of the families and in many countries associated with the support and the opportunities for academic and cultural exploration. For example, several studies presented higher rates of dropout in students with parents with lower levels of education. In the description the authors presented “responsibilities related to fatherhood, motherhood, or any other kinship that does not allow the student to devote him or herself entirely to his or her undergraduate studies” – this is also relevant to dropout but are not family background but Personal Variables, or Familiar Responsibilities.

The information related to the family context and how it affects the dropout of university students is expanded.

R1

Methodology. The same problems exist for 2.6, 2.7, 2.8, and 2.9. The variables presented are not clear and are mentioned in a simple way without reflecting the complexity and interaction between variables (as mentioned by the authors), which several times have mediating or moderating effects.

The definition of each of the dimensions is expanded.

R1

Methodology. It is not clear how the study occurs. Ethical procedures were not presented.

The discussion section has been expanded. The ethical component has been added.

R1

Methodology. How were data collection, data curation, and data analysis made? Which software was used?

Information on the statistical software used is included.

R1

Results. The results are presented in a very short way and do not address the constructs mentioned.

The results section is expanded with relevant information. The analysis is deepened in the discussion section.

R1

 Discussion and Conclusion. In this section is presented new information that could be presented in the introduction to better know the Latin-American, or Colombian, contexts.

Information from the discussion section of the Colombian context is included in the introduction section.

R1

References. Could be improved with more references; it is also important to include the titles of the manuscript for the language of the article.

More references are included

We look forward to your comments and hope to hear from you soon.

Thank you very much

_

The authors

Reviewer 2 Report

The issue of study is topical for many universities in the world.  The authors study  factors determining the decision to drop out of university students. The authors offer exploratory-descriptive field design with a participation of  372 students. The authors are focusing on the confluence of internal and external factors determining drop our rates. The study is contextualized by taking into account previous studies. Presentation of findings is coherent, balanced and compelling. Conclusions are supported by the results presented in the article. The concluding sentence in 2.2. section needs to be completed.

Author Response

February 22, 2023

Dear

Behavioral Sciences Editorial Office

Kind regards

According to the suggestions of our article by the reviewer, the following changes were made, properly marked with red letters in the article:

Reviewer

Comment

Response

R2

The concluding sentence in 2.2. section needs to be completed.

It is completed

R2

  In article does not appear: the purpose of the paper and the objectives of the paper.
It must be highlighted what is the purpose of this paper and the objectives you set out to achieve in this paper. The objectives of the study must appear in Introduction and in Methodology.

The research objective is included in the introduction and methodology.

R2

In line 139 the sentence is not finished.

It is completed.

We look forward to your comments and hope to hear from you soon.

Thank you very much

_

The authors

Reviewer 3 Report

The paper is interesting and studies a phenomenon with social and economic impact.

The work has many strong points, but it also has weaknesses that need to be improved.

My commnets for change:

1.     In article does not appear: the purpose of the paper and the objectives of the paper.

It must be highlighted what is the purpose of this paper and the objectives you set out to achieve in this paper.

The objectives of the study must appear in Introduction and in Methodology.

2.     In line 139 the sentence is not finished.

3.     Explain better what you wanted to convey in the paragraph in rows 199-205. It is not clear.

4.     In line 200, you are talking about the Varimax method. What is this method, what is its methodology? Please give explanations.

5.     How did the 372 students who participated in the study be selected? What sampling method was used? Please detail the method of selection of the sample and describe this sample (age, environment, from which universities the 372 students belong to, etc.) in the Methodology. Are they just students from Columbia University?

6.     How did you get the results (scores) from Table 2? Where do you get those values from? What does it represent? It is not enough that they are obtained with the help of the SPSS program, you need to explain what they represent. Please give explanations, be more clear.

7.     In row 215 is Table 3, not Figure 3.

8.     In the Conclusions please mention what are the limits of the research and who would benefit from this paper.

Hope to take my comments as a constructive advice.

Author Response

February 22, 2023

Dear

Behavioral Sciences Editorial Office

Kind regards

According to the suggestions of our article by the reviewer, the following changes were made, properly marked with red letters in the article:

Reviewer

Comment

Response

R3

Explain better what you wanted to convey in the paragraph in rows 199-205. It is not clear.

It contextualized what was intended to be done with the described process. The correlation between the dimensions and variables of the model is mentioned.

R3

In line 200, you are talking about the Varimax method. What is this method, what is its methodology? Please give explanations.

The method is explained.

R3

How did the 372 students who participated in the study be selected? What sampling method was used? Please detail the method of selection of the sample and describe this sample (age, environment, from which universities the 372 students belong to, etc.) in the Methodology. Are they just students from Columbia University?

The type of sampling used in the study is explained.

R3

How did you get the results (scores) from Table 2? Where do you get those values from? What does it represent? It is not enough that they are obtained with the help of the SPSS program, you need to explain what they represent. Please give explanations, be clearer.

It is explained that the results were obtained from an orthogonal rotation and a principal component analysis (factor loadings).

R3

In row 215 is Table 3, not Figure 3.

The information is organized

R3

In the Conclusions, please mention what are the limits of the research and who would benefit from this paper.

The limitations of the study and the beneficiaries of the findings are mentioned.

We look forward to your comments and hope to hear from you soon.

Thank you very much

_

The authors

Round 2

Reviewer 1 Report

The authors made clarify several parts of the manuscript, improving its overall quality.